# Cell Culture Models for Hepatitis E Virus

**DOI:** 10.3390/v11070608

**Published:** 2019-07-03

**Authors:** Rebecca Menhua Fu, Charlotte Caroline Decker, Viet Loan Dao Thi

**Affiliations:** 1Schaller Research Group at Department of Infectious Diseases and Virology, Heidelberg University Hospital, Cluster of Excellence CellNetworks, 69120 Heidelberg, Germany; 2Heidelberg Biosciences International Graduate School, Heidelberg University, 69120 Heidelberg, Germany

**Keywords:** hepatitis E virus, cell culture systems, stem cells, hepatocyte polarization

## Abstract

Despite a growing awareness, hepatitis E virus (HEV) remains understudied and investigations have been historically hampered by the absence of efficient cell culture systems. As a result, the pathogenesis of HEV infection and basic steps of the HEV life cycle are poorly understood. Major efforts have recently been made through the development of HEV infectious clones and cellular systems that significantly advanced HEV research. Here, we summarize these systems, discussing their advantages and disadvantages for HEV studies. We further capitalize on the need for HEV-permissive polarized cell models to better recapitulate the entire HEV life cycle and transmission.

## 1. Introduction

Hepatitis E virus (HEV) is a major causative agent of acute fulminant hepatitis [1]. The virus can be transmitted via the oral–fecal route, mainly through contaminated drinking water [1]. In some cases, the infection can also be blood-borne, for example, through blood transfusion [2]. It is found worldwide with the greatest prevalence in developing countries in South and East Asia. Hepatitis caused by HEV is usually self-limiting and clears in a few weeks in healthy individuals. However, the disease can be chronic in immunocompromised people and fatal in pregnant women or patients with previous liver diseases [3]. According to the World Health Organization (WHO), there was an estimated 44,000 deaths caused by HEV in 2015, which accounts for 3.3% of all deaths due to viral hepatitis (https://www.who.int/news-room/fact-sheets/detail/hepatitis-e). Therefore, this virus is considered a global health threat and significantly impacts the healthcare system.

The first recorded incidence of HEV dates back to 1978 in the Kashmir Valley of India, where a large-scale, water-borne epidemic of hepatitis spread to over 200 villages and resulted in 600 deaths in a seven-week period [4]. Infected patients tested negative for both hepatitis A and B virus (HAV and HBV), hence the viral entity responsible for the epidemic was first named enterically transmitted non-A, non-B hepatitis (ENANB) [5], and only later HEV. The virus is zoonotic and infects a broad range of organisms including pigs, wild boars, deer, and chickens. Among the diverse strains of HEV, there are at least four genotypes (GT, numbered 1 to 4) that infect humans, where GT1 and GT2 infect only humans, and GT 3 and GT4 are zoonotic [6]. In addition, there is also recent evidence of a single case of human infection by a new HEV genotype (GT7) found in camels [7], and two by rat HEV strains [8,9]. Since the discovery of HEV, the virus has been propagated in various cell lines (summarized in Section 3.1 and Section 3.2). However, most cell culture systems for HEV have been limited by low viral titer and slow viral replication. The lack of an efficient in vitro culture system is one of the biggest challenges in HEV research. As a result, off-label ribavirin and interferon-α remain the treatment of choice in chronic infections, but there are currently no drugs that specifically target HEV [10,11]. Therefore, it is important to develop efficient ways to propagate HEV in cell culture for the study of HEV biology and pathogenesis on a molecular level. Such systems can then be used to investigate viral protein functions, replication kinetics, entry mechanisms, and so on to screen for potential drug targets and identify new specific targets for therapeutic intervention.

This review summarizes existing cell culture systems for HEV, including hepatoma cells, primary hepatocytes, stem cell-derived hepatocytes, non-liver cells, and infectious cDNA clones of HEV. We will also discuss the need for polarized cell models to better recapitulate the entire HEV life cycle.

### 1.1. Genome Organization

HEV is an icosahedral, single-stranded RNA virus with a diameter of 27–34 nm [5,12]. The virus has a positive-strand RNA of 7.2 kb that is capped at the 5’ end and polyadenylated at the 3’ end. The HEV genome contains three open reading frames (ORFs). ORF1 encodes the domains responsible for genome replication: a methyltransferase (MeT), a papain-like cysteine protease (PCP), a helicase (Hel), and an RNA-dependent RNA polymerase (RdRp) domain (reviewed in the work of [13]). It is currently unclear whether these gene products are expressed as a single polyprotein or cleaved into individual proteins [14]. Viral replication also yields a bicistronic subgenomic RNA that encodes ORF2, the capsid protein, and ORF3, a protein found in the quasi-enveloped form of the virus that is involved in viral egress [15]. An additional ORF (ORF4) is exclusively expressed by HEV GT1 under cellular stress conditions and was shown to enhance the function of RdRp [16].

### 1.2. HEV Life Cycle and Transmission

The first HEV strain to be visualized by electron microscopy was from a stool extract and it was observed as a non-enveloped virus (nHEV) [5]. However, later studies showed that HEV particles released in cell culture supernatant are wrapped in host-derived membranes [17,18]. This allows the virus to bud in a non-cytolytic fashion, despite the absence of viral glycoproteins. The membrane-associated HEV particles are termed quasi-enveloped HEV (eHEV) [19].

The non-enveloped virions are ingested orally and penetrate the gut through a poorly understood mechanism [20]. They are then carried to the liver through the bloodstream where they infect, replicate, and propagate in hepatocytes (Figure 1). Viral progenies are released at both the apical and basolateral side of the hepatocyte, with eHEV found predominantly on the basolateral side and nHEV on the apical side [21] (Figure 2). There is evidence that the membrane associated with eHEV is derived from the trans-Golgi network, and that ORF3 interacts with proteins from the ESCRT machinery to direct budding of the virus into multivesicular bodies (Figure 1) [15,17,18].

It is speculated that when eHEV is released via the apical membrane into the bile duct, the quasi-envelope and ORF3 proteins are stripped off by bile acids followed by proteases in the duodenum [22,23]. The naked virus is more infectious than eHEV in vivo and in vitro [24,25,26], which might explain its presence in the feces, as it is the main route of HEV transmission. eHEV particles budding from the basolateral membrane into the bloodstream keep their quasi-envelope, which decreases cell attachment and thus entry [25]. Yet, it protects the particles from neutralizing antibodies [24,27], likely ensuring spread within the infected liver or to other host tissues [22], albeit with a low efficiency.

Because of the polarity of hepatocytes, different forms of HEV are secreted from different sides of the cell. Most existing HEV cell culture systems are based on cells grown in non-polarized monolayers and cannot recapitulate this aspect of the HEV life cycle as they support only poor virus spread. Attempts to investigate directional HEV infection as well as secretion using polarized cells on Transwells® have been made [21,28] and will be discussed later.

## 2. Viruses

### 2.1. HEV Patient Isolates

HEV isolates that were used to infect cells in culture are summarized in Table 1a. One of the first attempts to isolate HEV (formerly termed ENANB) from patients and propagation in cell culture was by Pillot and colleagues in the 1980s, where the virus was extracted from patient fecal samples from France and the Ivory Coast [29]. Hepatoma cell lines, PLC/PRF/5 and HepG2, as well as fibroblasts MCR5, were inoculated with antigen-positive stool samples, and HEV antigens were found in the culture supernatant of PLC/PRF/5 cells. Liver extracts from virus-positive monkeys also infected PLC/PRF/5 cells, thereby confirming the hepatotropism of the virus (Table 1a). In the following years, several other strains of HEV from Asia, South America, and Africa were characterized. These strains include GT1 Sar-55, 87A, and F23, which are still used today as infectious cDNA clones or as primary isolates [30,31,32,33,34].

After the discovery that HEV was the unique causative agent of acute hepatitis E, various attempts have been made to propagate HEV in cell culture. One of the most important studies to establish a cell culture system for HEV was conducted by Okamoto and colleagues, where 21 cell lines from a variety of organisms including humans, mice, rats, monkeys, cows, and dogs were inoculated with the JE03-1760F strain at a high viral load (2 × 10^7^ copies/mL) [56]. This strain was isolated from a Japanese patient who had an acute HEV GT3 infection. It replicated efficiently in 2 of the 21 tested cell lines: the hepatoma cell line PLC/PRF/5 and the lung cancer cell line A549. HEV RNA was detected in the culture supernatant eight days after inoculation and reached 10^8^ RNA copies/mL 50 days post-infection (dpi), similar to previous results obtained by Pillot et al [29].

In another study, Lorenzo et al. also showed virus spread in PLC/PRF/5 and A549 cells [57]. The JE03-1760F strain became detectable 12 dpi in PLC/PRF/5 cells and was successively propagated through at least thirteen generations of serial passaging. Another strain from GT4, HE-JF5, was also able to grow in PLC/PRF/5 and A549 cells to high titers after six passages (more details in Section 3.1). At day 10 post-infection, secreted ORF2 protein became detectable by Western blot analysis [53]. These results show that the virus adapts to growth in cell culture, and this can be used to develop an efficient cell culture system for producing HEV in vitro. Recently, another HEV GT3 strain, TLS 09/M0 (first described by the authors of [52]), was isolated by Capelli and colleagues. The authors used this strain to infect polarized F2 cells, a subclone of HepG2/C3A cells, and showed that F2 cells were even more readily infected by the TLS 09/M0 strain than parental HepG2/C3A cells [21].

However, up to this point, HEV in cell culture remained inefficient, and the zoonotic aspects of HEV still required further research. In an attempt to address these issues, Shukla et al. established a cell culture system for HEV in other host species [46]. HEV GT3 Kernow-C1 was purified from the feces of a patient chronically infected with HIV-1 and HEV. This strain infected not only the human hepatoma cell line HepG2/C3A, but also a broad range of animal cells including chicken, mice, and swine cells. In addition, the strain was found to be very efficient in infected HepG2/C3A cells after six passages, and was thus termed Kernow-C1/p6. Sequence analysis data showed an insertion of 58 amino acids in the hypervariable region (HVR) of ORF1 compared with other strains. The inserted sequence was identified to belong to the ribosomal S17 superfamily, which is conserved across species. Subsequent studies by the same group showed that this insertion was already present in the wild-type virus isolate to a small extent. It was selected in cell culture during the first passage, suggesting that the insertion conferred a significant growth advantage [58]. Of note, introducing the S17 sequence into an early passage of the Kernow-C1 strain enhanced its replication in hepatoma cells [58]. Likewise, the GT1 Sar55 strain was modified by introducing the S17 sequence from the GT3 p6 strain and also enhanced replication of the GT1 strain [38].

Interestingly, several other studies have shown evidence of similar insertions in the wild-type HEV viral genome that confer growth advantages in vitro. Nguyen et al. described an insertion of 39 amino acids from S19 ribosomal protein fused to the viral non-structural protein in the GT3 strain, LBPR-0379. The viral genome with the insert constituted only a minor species in the feces, but became the major species during passage in cell culture [43]. HEV strains with mutations and duplications in ORF1 have also been observed. Debing et al. identified point mutations in the RdRp region as well as a large in-frame 282-bp insertion in the HVR. Both the mutations and the insertion enhance viral replication and are associated with ribavirin treatment failure [59,60]. Finally, Johne et al. identified a rearrangement of the ORF1 region that involves a 116-nucleotide-long duplication and an insertion of 70 nucleotides derived from the 3′-terminal ORF1 region of the viral genome. This rearrangement also enabled the virus to replicate efficiently in cell culture [50].

In addition to fecal samples, HEV particles isolated from serum were also shown to be infectious in cell culture. In a study by Takahashi and colleagues, various strains of HEV GT 1, 3, or 4 were taken from patients during the acute phase of infection, which infected and replicated efficiently in PLC/PRF/5 and A549 cells. [27]. In a later study, Johne et al. confirmed these results, showing that another HEV strain from a serum sample named 47832 successfully infected and replicated in A549 cells [50]. In addition to A549 cells, Schemmerer et al. showed in a very recent study that the 47832 strain also propagated in PLC/PRF/5, HepG2/C3A, Huh-7 Lunet BLR, and MRC-5 cells [51]. In this study, the authors further isolated three HEV GT3 strains from patient serum: 14-16753, 14-22707, and 15-22016 [51]. These strains represent the predominant subtypes 3c, 3e, and 3f, respectively, which are currently circulating in Europe. These serum isolates replicated to high viral loads of 10^8^, 10^9^, and 10^6.5^ HEV RNA copies/mL at 14 dpi, respectively. In addition, they could persistently infect cell cultures with constant high viral loads (~10^9^ copies/mL) for more than a year. It is worth noting that unlike the Kernow C1/p6 and LBPR-0379 strains, these serum isolates do not have the ribosomal protein insertion in their genomes.

Serum isolates led to the discovery that infectious HEV particles also exist in blood, which was later found to be the quasi-enveloped form of HEV (Figure 1 and Figure 2). These particles are still infectious despite the presence of neutralizing antibodies in the serum because of the protective nature of the quasi-envelope. This is probably the reason that HEV infections can also be transmitted through blood transfusion [2].

In conclusion, fecal and serum HEV have been successfully isolated from infected patients and propagated in cell culture. Early studies from patient isolates confirmed the hepatotropism of HEV and led to the establishment of basic cell culture systems for HEV. However, there are still several drawbacks of using patient isolates in cell culture. For example, HEV particles from patient isolates are inefficient in spreading [28,36] (Table 1b), and variability between and within individual patients could lead to inconsistencies in experimental results.

### 2.2. HEV cDNA Clones

Propagation of HEV isolates from patients in cell culture improved the understanding and characterization of the virus. However, the variability mentioned above and the inability to genetically modify these strains were major drawbacks. Therefore, infectious HEV complementary DNA (cDNA) clones were developed to overcome these issues (Table 1a,b). The first attempt at making a cDNA clone was by Reyes et al. in 1990. Viruses isolated from the gall-bladder bile of cynomolgus macaques were used as a source to construct a cDNA library and the clone was named ET1.1 [61]. Despite the fact that a full-length cDNA clone was not constructed from this study, the hybridization experiments using ET1.1 as a probe provided evidence that the causative agent of ENANB was the unique viral entity HEV.

Subsequently, Panda et al. developed the first full-length cDNA clone from an epidemic isolate of HEV GT1 in India [62]. The three ORFs and the noncoding regions were cloned by subgenomic PCR amplification and the fragments were assembled by using different restriction enzymes. The culture supernatant from the RNA-transfected HepG2 cells using this cDNA clone was shown to be infectious in rhesus monkeys.

Following Panda and colleagues’ attempt, several other cDNA clones were developed. A study by Huang et al. reported the successful construction of infectious clones of swine HEV GT3 by reverse transcription-PCR of eight overlapping fragments spanning the entire HEV genome [63]. The RNA transcripts were transfected into Huh-7 cells and both ORF2 and ORF3 were detected. All three clones were replication competent and one of the clones was shown to successfully infect pigs when injected intrahepatically. Another GT3 cDNA clone based on strain G3-HEV83-2-27 was developed by Shiota et al. and the RNA of the infectious clone was successfully transfected into PLC/PRF/5 cells to study the C-terminal region of the capsid protein ORF2 [42]. Yamada et al. developed a full-length infectious cDNA clone from a fecal strain (GT3 JE03-1760F), and observed ORF2 expression when transfected into PLC/PRF/5 cells [45]. When cDNA-derived virus was inoculated into PLC/PRF/5 and A549 cells, they grew as efficiently as the fecal-derived virus.

A cDNA clone of the widely-used GT3 Kernow-C1/p6 strain was also constructed by Shukla et al., which was adapted to grow in HepG2/C3A cells and shown to be infectious in both human and swine cells [58]. In addition, a replicon of Gaussia luciferase-expressing HEV was shown to replicate more efficiently in two subclones of the PLC/PRF/5 cell line, namely PLC1 and PLC3 cells, than in the parental PLC/PRF/5 cells [64]. Finally, a cDNA clone of GT4 TW6196E strain was constructed, being replication competent in Huh-7 cells and infectious when inoculated into HepG2/C3A cells [54].

With the aim of developing animal models of HEV infection, rat HEV strains R63/DEU/2009 and pLAB350 were constructed. In vitro transcribed HEV RNA was injected into rats and virus particles from the feces were recovered [55,65]. They were shown to be infectious when inoculated into PLC/PRF/5 and Huh-7 cells. With the recent reports on humans contracting rat hepatitis E [8,9], these clones will surely be instrumental in assessing the zoonotic potential and mode of transmission of rat HEV to humans.

Finally, Huang et al. constructed an avian HEV cDNA clone that is replication competent in chicken liver cells (LHM cells) and infectious when inoculated into chicken livers [66].

As summarized in Table 1b, full-length cDNA clones replicate more efficiently in cell culture and result in higher reproducibility compared with patient isolates. They also allow for genetic modifications, which provide a good tool for the study of HEV protein functions. For example, mutations in the intergenic junction region [67], ORF2 [68], and ORF3 [69,70] can be achieved using cDNA clones to study the functions of these components. In addition, introduction of reporter proteins such as GFP or luciferase into cDNA clones to generate HEV replicons facilitated quantitative monitoring of HEV replication [35,58]. A very detailed and comprehensive overview on available HEV isolates and cDNA clones was also recently summarized by Meister and colleagues [71].

## 3. Cell Models for HEV Infection

### 3.1. Hepatoma Cell Lines

Since the discovery of HEV, propagation and production of HEV have been attempted in various cancer cell lines. Pillot et al. demonstrated that ENANB isolated from stool, now known as HEV, was able to infect PLC/PRF/5 cells [29]. Later, in 1997, Meng et al. used the same ENANB cell culture system to develop neutralization assays against HEV [72]. While these studies showed that HEV isolates can be successfully propagated in cell culture, virus replication level was very low in these cells. As stated previously, to develop an efficient cell culture system for HEV, Okamoto and colleagues screened 21 cell lines (including lung carcinoma cell line A549 as well as hepatoma cell lines HepG2 and PLC/PRF/5) for their ability to support the HEV life cycle [56]. As mentioned in Section 2.2, A549 and PLC/PRF/5 cells were shown to be infected in Okamoto’s study. Specifically, maintaining infected PLC/PRF/5 cells at 35.5 °C yielded the highest virus titer. In the aforementioned early studies from 1990s, the same culture system was used (inoculation of fecal suspension into PLC/PRF/5 cells), but the yield was never as high as in Okomoto et al.’s study. Okomoto and colleagues attributed this difference to the higher viral load in the inoculum used in their study. When they inoculated PLC/PRF/5 cells with fecal samples of low viral load, no HEV propagation was observed in PLC/PRF/5 cells. In another study, Tanaka and colleagues also developed an efficient culture system for GT4 HE-JF5/15F strain in PLC/PRF/5 and A549 cells [53]. A fecal sample with an HEV load of 1.3 × 10^7^ copies/ml was inoculated into PLC/PRF/5 cells and the viral load reached 2.8 × 10^6^ copies/ml at 60 dpi. However, the viral load increased significantly and HEV RNA appeared significantly earlier after successive passages. After six passages in A549 cells, the HEV progenies reached 3.9 × 10^8^ copies/ml in the culture supernatant at 20 dpi. In addition, Takahashi et al. showed that HEV GT1 can also be cultured in PLC/PRF/5 cells and passaged in A549 cells [27]. Interestingly, subclones of A549 and PLC/PRF/5 cells support HEV replication with different efficiencies [51,73].

Shukla et al. found that the semi-purified HEV GT3 Kernow-C1 viral strain from feces infected five human and one rhesus monkey cell line (including HepG2/C3A, Huh7.5, PLC/PRF/5, and A549 cells). Infected cells were found in all six cultures, with HepG2/C3A cells being the most permissive (7.5-fold higher foci count) [46]. However, similar attempts to adapt Kernow-C1 to grow in A549 and PLC/PRF/5 cells were unsuccessful. In addition to human hepatoma cells, HEV GT3 strains isolated from pigs and deer were also shown to be infectious in mouse, chicken, and deer liver cells, though the permissiveness of the animal cell lines for HEV was significantly lower than that of human hepatoma cells [46].

### 3.2. Extra-Hepatic Manifestations and Non-Hepatoma Cell Lines

Despite being a hepatotropic virus, studies have shown extra-hepatic manifestations associated with HEV, including renal and neurological symptoms [74,75]. Negative-strand HEV RNA was also detected in colons, small intestines, and lungs of infected pigs, indicating HEV replication in these organs [76].

One of the early HEV studies showed that the virus was able to infect 2BS cells, a human fetal lung diploid fibroblast cell line, as well as A549 cells [40]. HEV strains isolated from the feces of four patients with acute hepatitis E were inoculated into various cell lines including 2BS and A549 cells and both were susceptible to all HEV strains used. In another study, Huang et al. passaged and propagated HEV strain 87A GT1 in 2BS cells and inoculated the culture supernatant into A549 cells, resulting in cytopathic effects and detection of viral RNA [41].

Interestingly, there is evidence that when inoculated with the same multiplicity of infection (MOI), virus progeny appears earlier in A549 cells than in PLC/PRF/5 cells, while the latter tend to generate higher viral loads at later stages [56]. The same study also showed that the virus titer used in the initial inoculum has an effect on HEV replication efficiency in A549 cells compared with PLC/PRF/5 cells. Inoculation at a higher MOI resulted in higher replication efficiency in PCL/PRF/5 cells than A549 cells, while a lower MOI resulted in more efficient growth of the virus in A549 cells with an earlier appearance of HEV progenies in the supernatant. This phenomenon may be because of the fact that A549 cells are more immune competent than PLC/PRF/5 cells, as hepatoma cells in general are defective in their innate immune response. Therefore, a lower MOI would be better in A549 cells, as too much virus is likely to induce an inhibitory innate immune response in these cells. As A549 cells are highly permissive to HEV, it is worth investigating whether the lung is a true replication site of HEV.

In a study to investigate cross-species infections of cultured cells by HEV, Shukla et al. showed that the fecal virus of the Kernow-C1 strain was able to infect three pig kidney cell lines (LLC-PK1, LLC-PK1A, and SK-RST) [46]. A few foci were also observed in dog (MDCK), cat (CRFK), and rabbit kidney cells (LLC-RK1). It is interesting that the Kernow-C1 strain used in this study infected a wide range of host species, which could be attributed to the generation of a complex quasi-species during prolonged infection in an immunocompromised patient host.

During its life cycle, HEV is shed in the feces and enters the human body through the intestine [77]. Consistent with this observation, a few studies have shown that HEV is also able to infect intestinal cells in culture. Upon transfection of GFP-tagged HEV transcripts of the GT1 Sar-55 strain into Caco-2 cells (a human intestinal cell line), the observed GFP signal was comparable to hepatoma cell lines PLC/PRF/5 and Huh-7, indicating successful virus replication in Caco-2 cells [35]. In another study by the same group, the same HEV strain was shown to replicate in polarized Caco-2 cells, where the virus was released at the apical membrane [28]. Similarly, Shukla et al.’ s study, mentioned earlier, observed foci in Caco-2 cells infected with HEV, in addition to the human liver cells and animal kidney cells [46].

Studies have shown that HEV GT1 is more severe and dangerous in pregnant women, but the underlining mechanisms remain a mystery [78]. Interestingly, a study by Steinman’s group showed that HEV GT1 and 3 are able to replicate in human placental cell lines JEG-3 and BeWo [37]. HEV replication kinetics in JEG-3 cells were similar to those observed in HepG2 cells. JEG-3 cells also supported HEV assembly, release, and production of infectious particles. In addition, this study showed that HEV replication in placenta cells is inhibited by ribavirin. As ribavirin is contraindicated in the case of pregnancy, placental cell models for HEV could be used to develop novel drugs against HEV for pregnant women.

HEV infection is known to be associated with neurological disorders including Guillain–Barré syndrome, polyradiculopathy, and neuralgic amyotrophy [75]. Although direct evidence of HEV infection in the brain is lacking, systems for culturing HEV in neuronal cell lines have been developed. One of the first attempts to culture HEV in neuronal cells was by Drave and colleagues, where five neuronal cell lines were transfected with a GT3 replicon based on the Kernow-C1/p6 strain. The virus replicon was shown to be replication competent in all five cell lines. In one of the cell lines (oligodendrocytic cell line M03.13), HEV replication was even as efficient as in HepG2 cells [38]. The M03.13 cells also supported replication of the modified HEV GT1 strain Sar55 pSK-E2/S17. In another study, Zhou et al. similarly showed successful infection of HEV in neuronal cell lines. Cell culture-derived HEV Kernow-C1/p6 was inoculated into four neuronal cell lines and two of them were as susceptible to HEV as Huh-7 cells. In addition, they showed that HEV was also infectious in human induced pluripotent stem cell-derived neuronal cultures [79]. However, another study by Helsen et al. showed that stem cell-derived neuroprogenitors only supported HEV replication upon transfection with a HEV subgenomic replicon, but not infection with the full-length virus [80]. The exact reasons for these conflicting results are not known. The neuroprogenitor cells used in the two studies were from different sources, which may have contributed to the discrepancy in their results.

### 3.3. Primary Cells

Cancerous cell lines are available, while being easy to culture and genetically manipulate, but their transformed nature restricts relevant studies of, for example, innate immune responses (Table 2). In particular, most hepatoma cell lines lack phase I, II, and III drug metabolizing enzymes, which makes them unsuited for the assessment of anti-viral treatments [81]. The most authentic culture system are primary cells, which have an intact genome and thus expression profile that resembles the cells in the original organ most closely. Moreover, primary cells derived from different organs allow the study of extrahepatic manifestations of HEV infection. Yet, primary cells usually show a high variability, which depends on the donor’s health status, the isolation protocol, and the culture conditions. Additionally, their availability is limited, especially in the case of primary human hepatocytes (PHHs), which are the most relevant target cells for HEV studies. In addition, PHHs often de-differentiate after isolation because of changes in three-dimensional architecture, as well as injuries and pro-inflammatory signaling during the isolation process. They do not only show reduced hepatic function, but also usually die within a few days after isolation when cultured as monolayers (reviewed in the works of [81,82]).

In a first attempt, Tam et al. cultured primary hepatocytes isolated from HEV-infected cynomolgus monkeys for more than two months and showed that HEV can replicate in long-term culture [83]. Later, the same group published a protocol describing the isolation of primary cynomolgus monkey hepatocytes and infection with the HEV GT1 Burma strain [84]. In these two studies, HEV RNA of both strand polarities was found in infected cells and genomic RNA was detected in cell culture supernatants.

Clinical strains of HEV GT1 and GT3 were used to infect ex vivo transplants of maternal decidua and fetal placenta. The production of infectious progeny virus and resulting tissue damage was greater for HEV GT1 compared with HEV GT3 [85].

Swine-derived HEV GT3 and 4 isolates were used to infect PHHs [49]. Interestingly, the number of cells in a foci was found to increase with time after infection, indicating that HEV spread in PHH was rather the result of cell-to-cell transmission through the cell membrane than infection of HEV through the culture media [49].

The widely-used Kernow-C1/p6 strain can also infect immune competent PHHs [47] and human fetal liver cells [36] (Table 1a). The Kernow-C1/p6 strain can further infect, as mentioned above, primary mouse neurons [79]. Moreover, Kernow-C1/p6 RNA was used to transfect mouse embryonic fibroblasts [86], as well as murine and human primary liver organoids [87]. The observation that the p6 strain may be able to infect and replicate in cells of murine origin is in agreement with its widened host range; yet, some results obtained with this particular strain must be taken with caution, as discussed below.

### 3.4. Stem Cell-Derived Models

Hepatocyte-like cells (HLCs) differentiated from human-induced pluripotent stem cells (hiPSCs) or embryonic stem cells (hESCs) represent a valuable alternative to PHHs, as they are not limited by some of the drawbacks PHHs pose [88,89] (Table 2). Established stem cell lines are permanently available and can thus yield reproducible results. Although not being fully mature, the HLC’s expression profile and metabolic function resemble adult hepatocytes more closely than hepatoma cell lines [90]. Further, HLCs can be cultured over long time periods (up to one month)*,* while surviving better than PHHs [91]. This is critical for assessing the effect of chronic virus infection on the cell. However, the expression levels of hepatocyte markers change slowly over time [91,92], which should be monitored for relevant markers during long-term studies. Another major advantage of the system is that iPSCs can be induced from human samples [93] and differentiated into patient-specific HLCs to create personalized HEV infection models [22]. In addition, genetic manipulation of iPSCs, for example, by CRISPR/Cas9 [94] or viral transduction [95], is possible to modulate host factors and obtain HLCs with a desired phenotype.

Yet, we have to keep in mind that HLC differentiation remains time-consuming and complicated. HLCs are more physiologically relevant, but they retain an immature phenotype that cannot fully recapitulate all hepatocyte functions (reviewed in the work of [90]). Likely, differentiation under three-dimensional (3D)-culture conditions may improve this and yield HLCs that resemble PHHs more closely. Efforts in this direction are underway, as discussed in the next chapter.

Stem cell-derived HLCs have been used to study hepatitis viruses and other hepatotropic infectious diseases. For example, HLCs support infection with HAV [96], HBV [91,97,98], and hepatitis C virus (HCV) [88,99,100,101], as well as Dengue [102,103] and Zika virus [104]. HLCs were also shown to be permissive for different *Plasmodium* species, including *P. falciparum* [105].

We and others have shown that HLCs support the full replication cycle of the cell culture-adapted HEV GT3 Kernow-C1/p6 strain [80,106]. Moreover, we also provided evidence that non-adapted patient isolates of HEV GT1-4 can infect HLCs and replicate to high levels [36]. HLCs thus constitute an important tool for understanding HEV biology, especially with regards to the investigation of HEV GT2 strains, which, to our knowledge, do not replicate in hepatoma cells [22]. 

Similarly, HCV [99] and HBV [97] clinical isolates can infect HLCs. These viruses do not infect hepatoma cells, unless they ectopically express critical host factors, such as SEC14L2 for HCV [107] and the sodium-taurocholate cotransporting polypeptide (NTCP) for HBV [107]. The common denominator in these observations are the hepatoma cells, which, because of their transformed nature, likely lack essential host-factors of virus replication. To overcome this, isolates were adapted to efficient growth in hepatoma cells, as discussed in Section 2.1. During this adaptation, viral genomes with single point mutations or even insertions, which may alter the biology of the virus, become the dominant species. For example, disruption of cyclophilin A (CypA) or treatment with the CypA-inhibitor cyclosporine A, an immunosuppressant, enhanced replication of the HEV GT3 Kernow-C1/p6 strain in hepatoma cells and HLCs [36,108], but had no effect on non-adapted isolates of HEV GTs 1-4 in HLCs [36]. Similarly, two other immunosuppressants had no effect on sustained virologic response in ribavirin-treated patients [109], whereas in cell culture, HEV GT3 Kernow-C1/p6 replication was inhibited by mycophenolic acid [108] and enhanced by mTOR inhibitors [110].

Importantly, when culturing swine-derived HEV GT3 and 4 isolates in PHH (as mentioned in Section 3.3), the authors did not find any differences between inoculated and propagated HEV [49]. Therefore, it is critical to confirm findings made with adapted HEV viruses with non-adapted isolates in a physiologically relevant cellular system, such as PHHs or HLCs.

## 4. Polarized Cell Models for HEV Infection Studies

Hepatocytes, like all epithelial cells, act as an important barrier between the body and the outside world. In order to exert their barrier function while providing uptake of nutrients (intestine) or oxygen (air), epithelial cells are polarized. Hepatocytes stand out through their complex multipolar architecture (Figure 2A). Each cell is in contact with multiple neighboring hepatocytes and faces at least one blood vessel through fenestrated endothelium via its basolateral sides. In addition, each hepatocyte has at least one apical domain, which faces a bile canaliculus. The canaliculi are often bounded by only one or two hepatocytes. Their particular polarization is vital for hepatocytes to properly execute their functions, which include canalicular bile secretion via their apical membranes, while mediating the uptake and secretion of serum proteins into the bloodstream via their basolateral membranes (reviewed in the work of [111]). The hepatocyte’s cell polarity, physiology, and function strongly depend on cell–cell and cell–extracellular matrix (ECM) interactions that are provided in the 3D tissue environment of the liver.

As described in the introduction, the HEV life cycle and transmission highly depend on hepatocyte polarization: HEV enters hepatocytes at their basolateral side, and progeny virions are mainly secreted from their apical side [96]. HEV ORF3 protein was shown to accumulate at the apical side of hepatocytes in HEV-infected humans [112] and in liver-chimeric humanized mice [26]. Investigations of the mechanism and determinants of directional HEV secretion are now possible with the recent efforts in developing polarized hepatocyte culture systems (reviewed in the work of [81,113]), which will be summarized in the following paragraphs.

### 4.1. Polarized Hepatocytes without Access to Both Domains

Different approaches have been developed to induce, keep, and/or restore the complex polarization of hepatocytes (Figure 2A) in cell culture. The bipotent HepaRG cell line can be differentiated into hepatocyte-like and biliary cells, which self-organize to form two-dimensional bile canaliculus-like structures [114]. HepaRG cells support the replication of swine-derived HEV GT3 [115] and the HEV GT3 Kernow-C1/p6 strain [116]. Similarly, other hepatoma cell lines were grown under conditions favoring the formation of polarized, three-dimensional structures, such as, for example, HepG2 cells [117]. Several of these hepatoma-based spheroid models have been developed for the study of HCV: Huh-7 cells in matrigel [118], Huh-7.5 cells in galactosylated cellulosic sponges [119] or polyethylene glycol-based hydrogels [120], HuS-E/2 cells grown in a thermoreversible gelatin polymer [121], and FLC4 cells cultured in a radial-flow bioreactor [122]. For HEV, a 3D-polarized system of PLC/PRF/5 cells grown on porous microspheres in a rotating vessel system was shown to support the full virus life cycle. Of note, inoculation of PLC/PRF/5 cells cultured as a monolayer with the same swine-derived HEV GT3 strain, NLSWIE3 [123], did not lead to productive infection and replication [48].

Although cancer cell lines show better hepatic protein expression in complex cultures compared with monolayers, they still retain their transformed phenotype [81], which limits their use for authentic hepatotropic viral studies.

Culturing primary hepatocytes in sandwich monolayers or in spherical liver organoids helps in maintaining their complex polarization, which can be further improved by co-culture with non-parenchymal cells [81,124]. These systems have been applied for the studies of HBV, HCV, and malaria [97,125,126].

HLCs can also be cultured under polarization-favoring conditions, which improve their differentiation status and thus hepatocyte functions. For example, growing iPSC-derived HLCs in micropatterned co-culture with murine embryonic fibroblasts yielded HLCs with a more matured phenotype and enhanced longevity compared with HLCs kept in sandwich monocultures [127]. Self-organizing, polarized HLCs grown as spheroids can be obtained by culturing cells on very low binding surfaces [128,129] or embedded in hydrogels [130]. Recently, a complex stem cell-based liver organoid system based on differentiating hiPSC-derived endoderm, together with human umbilical vein endothelial cells and mesenchymal stem cells in microwells [131], was used for HBV infection studies [132]. The liver organoids supported HBV replication for 20 days and secreted more viral DNA from 7 dpi compared with non-polarized HLCs, which the authors differentiated from the same hiPSC-endoderm [132].

Three-dimensional culture models of cancer cell lines, primary hepatocytes, or stem cell-derived HLCs represent promising tools to study HEV biology. Yet, directional studies are restricted, as these complex systems do not provide access to the apical domains, which face the closed canaliculi (Figure 2A).

### 4.2. Polarized Hepatocytes with Access to Both Domains

To discriminate between events that take place at either the apical or basolateral domain of polarized cells, a system in which both sides are clearly separated and accessible would be ideal. In particular, the directional formation and secretion of quasi-enveloped or naked HEV progenies from either hepatocyte membrane are not well understood. Furthermore, the role of bile acids and other host factors that may be important for the maturation of infectious HEV particles is not well studied.

For directional HEV studies, it would be useful to have hepatocytes with a columnar polarization (Figure 2B), similar to lung or intestinal epithelial cells. Columnar polarization can be achieved by culturing epithelial cells on a microporous membrane that is coated with ECM. Different media applied to both sides of the cells can mimic the directional flow to further support polarization. Once polarized, epithelial cells form tight junctions that separate the two sides [133], allowing side-specific virus infection and separate analysis of both apical and basal supernatants.

Immortalizing cell lines is associated with cell transformation that often involves the loss of epithelial cell polarity, which is why only a few existing cell lines allow for investigations into the effects of cellular polarity [111]. Among these, Caco-2 cells columnar polarize when grown on Transwells® and were used to study directional HAV and HEV infection [28,134]. Progeny virus was preferentially secreted from the apical side in both cases. Similar to observations in the liver, HEV ORF3 protein was mainly localized at the apical membrane of polarized Caco-2 cells [28]. Yet, in order to reach the sinusoidal bloodstream and eventually the liver, HAV and HEV should be ideally secreted from the intestinal cell’s basal membrane. A revised comparison with non-cancerous polarized intestinal epithelial cells such as hESC/iPSC- or adult stem cell-derived intestinal organoids (reviewed in the work of [135]) or even polarized primary intestinal cells would be favorable.

As mentioned before, plated PHH quickly dedifferentiate and lose their polarity. Their complex polarity can be sustained or restored by culturing them in sandwich cultures, but they cannot, to our knowledge, acquire a columnar polarization. Likewise, common hepatoma cell lines cannot be columnar polarized, owing to reasons mentioned above [111]. Only a few subclones have been identified and were used to study vectorial trafficking of HAV, HBV, and HCV. For example, the N6 subclone of the HepG2 cell line [136] can be grown as a columnar, polarized monolayer on Transwell® membranes, and was shown to support polarized trafficking of HAV [134,136] and HBV [137]. Similarly, Belouzard et al. described two Huh-7 subclones (15 and 1SC3) that can be polarized on Transwells® and support side-specific entry and secretion of HCV [138].

Recently, Capelli et al. identified a HepG2 subclone (F2) that can be columnar polarized on Transwells® and that is highly permissive for HEV GT1 and GT3 isolates [21]. They demonstrated that the infected cell clone secretes HEV particles in a side-specific manner, with a large majority released apically. The directional apical secretion was similar for both genotypes and was not dependent on whether the cells had been infected with nHEV or eHEV. The particles secreted to the culture medium on both sides were quasi-enveloped, but of slightly different densities. Transcytosis of nHEV or eHEV through the polarized cells was not observed [21]. So far, this has been the only published culture model describing studies of vectorial HEV trafficking.

Some evidence suggests that hepatocyte differentiation involves columnar intermediates [111]. To overcome some of the hurdles hepatoma cells and PHHs pose, we recently developed a stem cell-based differentiation protocol to generate columnar polarized HLCs [96]. During differentiation, the cells were grown on Transwell® membranes and fed with different media from the basal and apical sides. Polarized HLCs were permissive for the HEV GT3 Kernow-C1/p6 strain and progeny virus particles were secreted from both sides. Apically released HEV was mainly non-enveloped, whereas basolaterally released HEV was mainly quasi-enveloped (Figure 2B). The discrepancy between quasi-enveloped particles released apically from polarized HepG2 cells [21] and naked particles in the apical supernatant of polarized HLCs [96] might be the result of the different nature of the cell types used. Similar to HEV, Hirai-Yuki et al. observed only quasi-enveloped HAV particle release from either apical and basolateral membrane of infected polarized HepG2-N6 cells [134]. The particles were only converted into non-enveloped particles when treated with high concentrations of exogenous human bile salt. This is likely because of defects in bile salt synthesis [139] and other cellular pathways in HepG2 cells. Although Capelli et al. found bile salts in the supernatants of their HepG2-F2 cells [21], the local concentrations may be too low and/or additional factors may be missing in their model, such as, for example, a protease that could be necessary to efficiently strip off the lipid envelope of HEV particles.

## 5. Conclusions and Remaining Questions

Recent improvements in cell culture systems have significantly advanced HEV research. In this review, we discussed their advantages and disadvantages (summarized in Table 1 and Table 2). The development of infectious clones, identification of permissive cells, and optimization of HEV growth were major breakthroughs that advanced molecular studies of HEV. These achievements can be further complemented by efforts into improving hepatocyte systems that are physiologically more relevant, but at the same time accessible, convenient, and highly permissive for HEV infection. Current cell culture models only poorly support HEV spread. Therefore, future systems ideally recapitulate cell polarity and co-secrete bile acids and/or other host-factors to fully mature HEV particles into their non-enveloped form. These types of systems will likely better support virus spread and thus the entire HEV life cycle. Ideally, these systems will also support infection with clinical isolates, as cell culture-adapted HEV responds differentially to treatment with anti-viral drugs compared with non-adapted isolates. 

We do believe that stem cells can be valuable key players here. Their potential to give rise to different cell types facilitates not only studies of HEV cell tropism, but also the creation of isogenic, complex co-culture systems that better mimic the multicellular environment in the liver. Further, the plasticity of stem cells opens up the possibility to create differentiated cells with a desired polarity, as we have done to generate columnar polarized HLCs [96]. Both the cell polarity as well as the presence of other cell types may further mature HLCs to bring them closer to PHHs. Finally, HLCs are highly permissive for infection with HEV isolates [36].

Many questions about HEV remain. Each step of the viral life cycle remains poorly characterized and host factors regulating HEV infection are mainly unknown. How do nHEV particles enter the host cell? Is viral entry and thus spread also possible at the apical hepatocyte surface? Where in the cell does HEV genome replication and the assembly of progeny particles take place? Are nHEV and eHEV differentially released, and how is the preferential release to the bile regulated? How can HEV GT3 and 4 infections persist? What determines the tissue and species tropism of HEV? Future research with improved cell culture systems may help to clarify these questions. These efforts will lead to a better understanding of HEV biology, which is crucial for identifying new drug targets and testing anti-viral treatment strategies.

## Figures and Tables

**Figure 1 viruses-11-00608-f001:**
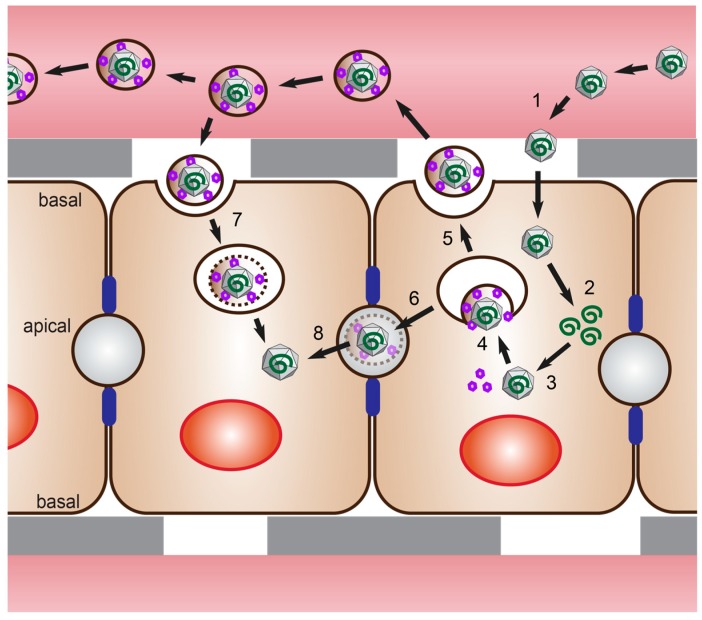
Hepatitis E virus (HEV) life cycle and transmission. **(1)** Supposedly naked HEV (nHEV) particles reach the liver and infect hepatocytes through their basolateral membrane. **(2)** Replication of the HEV genome (green) mediated by HEV nonstructural proteins occurs in a yet uncharacterized cellular compartment. **(3)** Translation of the subgenomic RNA leads to the translation of open reading frame 2 (ORF2) capsid protein (grey) and ORF3 protein (purple). The assembly site of infectious HEV particles is unknown. **(4)** Progeny HEV particles bud into multivesicular buddies mediated by the interaction of ORF3 with the ESCRT machinery of the host cell. **(5)** Basolaterally secreted HEV particles are quasi-enveloped (eHEV) and circulate in this form in the bloodstream. **(6)** Apically secreted particles supposedly bud as eHEV particles and are then matured into nHEV particles. While exact modes of nHEV entry (1) are unknown, **(7)** eHEV particles enter through clathrin-dependent endocytosis. Cell-to-cell transmission or **(8)** transmission between neighboring apical domains have not been described yet.

**Figure 2 viruses-11-00608-f002:**
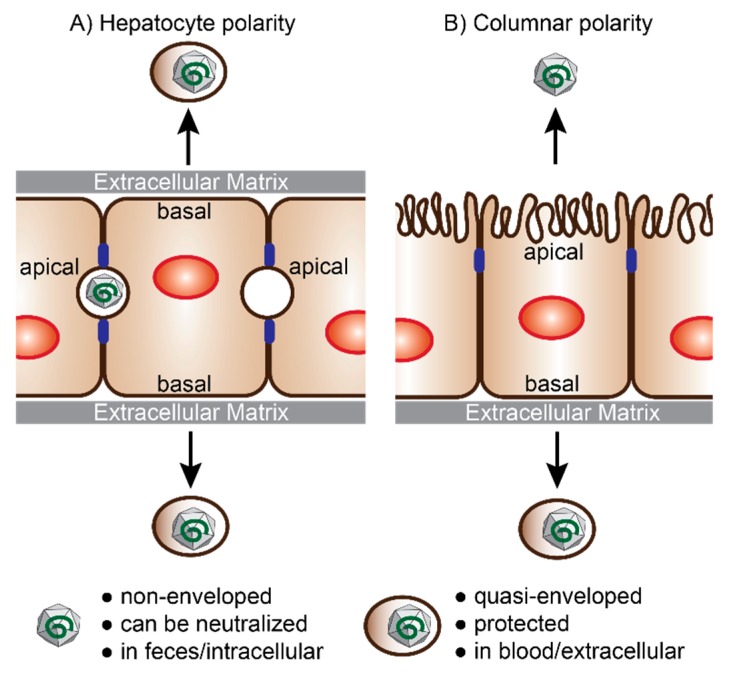
Cell polarity and hepatitis E virus (HEV) secretion. HEV particle release from hepatocytes with (**A**) complex hepatocyte or (**B**) columnar polarity. Blue boxes are tight junctions separating apical and basal membranes. Adapted from the work of [22].

**Table 1 viruses-11-00608-t001:** Summary of viruses. (a) Hepatitis E virus (HEV) isolates and cDNA clones. HEV strains applied in cell culture and the corresponding susceptible cells. If not stated otherwise, the cells are of human origin. PHH = primary human hepatocytes, HLCs = hepatocyte-like cells, ORF = open reading frame, wt = wild-type. (b) Comparison of HEV patient isolates with cDNA clones.

(**a**)					
**Genotype**	**Strain**	**cDNA Clone**	**Adaptation in Cell Culture**	**Susceptible Cells**	**Ref**
ENANB		Yes		PLC/PRF/5	[29]
1	Sar-55	Yes		Huh 7	[28]
				PLC/PRF/5	[35]
				Caco-2	[28]
				HLCs	[36]
	Sar55/S17	Yes		BeWo	[37]
			Introduced S17 sequence from Kernow-C1/p6	JEG-1	[37]
				M03.13	[38]
	F23			PLC/PRF/5	[39]
	87A			2BS	[40,41]
				A549	[40,41]
2	MEX-14			HLCs	[36]
3	HEV83-2-27	Yes		PLC/PRF/5	[42]
	LBPR-0379	Yes	Insertion of 39 amino acids from S19 ribosomal protein in wt virus, selected in cell culture	HepG2	[43]
	JE03-1706	Yes	Gained 13 mutations after 10 passages	PLC/PRF/5	[44]
				A549	[45]
	Kernow-C1	Yes	S17 insertion and 54 synonymous mutations in wt virus, selected after six passages in cell culture	Hep G2	[46]
				Huh-7	[46]
				SH-SH5Y	[38]
				SK-N-MC	[38]
				U97	[38]
				U343	[38]
				M03.13	[38]
				BeWo	[37]
				JEG-3	[37]
				LLC-PK1 (swine)	[46]
				OHH1.Li (deer)	[46]
				MDCK (dog)	[46]
				CRFK (cat)	[46]
				LLC-RK1 (rabbit)	[46]
				CMH (chicken)	[46]
				Hepa 1-6 (mouse)	[46]
				PHH	[47]
				HLCs	[36]
	NLSWIE3			PLC/PRF/5	[48]
	swJR-P5			PHH	[49]
	swJB-E10			PHH	[49]
	swJB-M8			PHH	[49]
	US-2			HLCs	[36]
	47832		Insertion in the hypervariable region of ORF1 in wt virus, gained 25 point mutations after two passages	A549 PLC/PRF/5 HepG2/C3A Huh-7 Lunet BLR MRC-5	[50]
					[51]
					[51]
					[51]
					[51]
	14-16753			A549 PLC/PRF/5 HepG2/C3A Huh-7 Lunet BLR	[51]
					[51]
					[51]
					[51]
	14-22707			A549 PLC/PRF/5 Huh-7 Lunet BLR	[51]
					[51]
					[51]
	15-22016			PLC/PRF/5 HepG2/C3A Huh-7 Lunet BLR	[51]
					[51]
					[51]
	TLS 09/M0		A 75-nt insertion in the polyproline region after 48 months of infection in vivo	F2 (subclone of HepG2/C3A)	[52]
4	HE-JF5		Gained 10 mutations after six passages	PLC/PRF/5 A549	[53]
	TW6196	Yes		HepG2	[54]
	swJB-H7			PHH	[49]
Rat	R63/DEU/2009 8	Yes	Gained 9 mutations after two passages	PLC/PRF/5	[55]
(**b**)					
			**Patient Isolates**	**cDNA Clones**	
Replication in cell culture			+	++	
Replication in animal models			++	+	
Physiological relevance			+++	+	
Pan-genotype			+++	-	
Reproducibility			-	+++	
Genetic manipulation			-	+++	

**Table 2 viruses-11-00608-t002:** Comparison of available cell systems for HEV research.

	Cancer Cells	Stem Cell-Derived Cells	Primary Cells
Availability	+++	++	-
Reproducibility	+++	++	-
Genetic modification	+++	++	-
Physiologically relevant	-	++	+++
Pan-genotype	-	++	+++
Cellular polarity	+	++	++

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
