# Peer review of "Cell Culture Models for Hepatitis E Virus"

_viruses, 2019, doi:10.3390/v11070608_

Round 1
Reviewer 1 Report
The efficient cell culture models for HEV are urgently needed for HEV research. The authors here summarized the cell culture models for HEV and this review is very helpful for the readers to know the history and current status about HEV cell culture models. The authors also highlight the importance of polarized hepatocytes for understanding of HEV life cycle and transmission. Meanwhile, the authors also describe the advantages and disadvantages of some cell models which are very insightful to the readers. However, there are some issues I concern. Firstly, the authors do not cite the previous studies in some important points, and I will specify some of them blow, but not limited to these. Secondly, I think the authors pay too much attention to the polarized hepatocytes cell culture models; this review is not specifically for the polarized hepatocytes model, I would like to suggest the authors to consolidate this section. Thirdly, across the manuscript, the authors use four unpublished data to support their points in the review, which cannot be accepted. Fourthly, some description is not correct or not presented in a logical way, and I specify some points as follow.
Some specific issues are here:
“Hepatitis E virus (HEV) is the causative agent of acute fulminant hepatitis E.” I cannot agree with the authors, because acute not necessarily fulminant, or chronic hepatitis E is also caused by HEV.
P1, L31: Leave two blank spaces at the beginning of this paragraph which should be consistent with other paragraphs.
P1, L22 and 38: Relevant reference(s) is missing.
P2, L54: if there is not 1.1 section, there should not have 1.2 section here.
P2, L57-61: It is still controversial that ORF1 function as a single polyprotein or cleaved into individual proteins. The authors cannot mislead the readers as they described in the manuscript.
P3, L89: “in” and “within” cannot be used together here.
P3 L100. PLC/PRF-5 and HepG2 cells are hepatoma cell lines, not hepatoma lines. The authors should check other places across the manuscript to use the correct way to describe the cells.
P3, L108, Citation.
P3 L115, Lung cancer cell line A549, same as my previous point 7.
P5, L197, know-C1 GT3/p6 is not correct and not consistent in the manuscript. It should be Know-C1/p6 (GT3).
P7, L254: Relevant reference(s) is missing after “cells”
P9 L325, “manipulated” instead of “manipulate”.
P9 L353, Avoid to use “famous” here. The authors can use “wide-used” to replace the “famous”.

Reviewer 2 Report
The authors nicely recapitulate past and current HEV cell culture systems. Although a recent excellent review on HEV cell culture systems by Todt et al. (2019 Antiviral research) has been published, the authors of this review present a slightly different and –in terms of models– wider perspective. Not only immortalised, primary and hepatocyte-like cells are described but also recent efforts to generate polarized models to better study the HEV life cycle. Moreover, they emphasize on the importance of confirming findings made with cell culture-adapted HEV strains.
Therefore, this review will be helpful for the community and I only have minor suggestions listed below.
Minor comments:
Line 22 and 85,86:
The statement that HEV transmitted mainly through contaminated drinking water certainly holds true for genotype 1. However, HEV genotype 3 and 4 are mainly transmitted alimentary and to a lesser extent via blood products.
Line 32:
According to reference [2], the number of 1,700 deaths correspondes to the total number of deaths occured in Kashmir valley accross 10 periodic epidemics until 2013. This should be clarified.
Line 74:
Could the authors elaborate on why they think only non-enveloped virions are ingested orally? Genotypes 3 and 4 are probably mainly ingested (or transfused) as quasi-enveloped virions.
2.1 HEV patient isolates
The authors summarize HEV patient isolates in Table 1a. However, two recently published studies by Capelli et al. (2019, J. Virol.) and Schemmerer et al. (2019, Viruses) are missing. The authors should consider adding these HEV strains (TLS-09/M0 (initially described by Lhomme et al. 2014, J. Virol.), 14-16753, 14-22707 and 15-22016) to Table 1a and section 2.1 HEV patient isolates.
Moreover, strain 47832 was also propagated in PLC/PRF/5, HepG2/C3A, HuH-7 Lunet BLR and MRC-5 (Schemmerer et al., 2016 Viruses).
Line 107:
Did the authors want to reference to the isolation of strains Sar-55, 87A and F23? If so, would references [24], [Huang et al. 1992, J. Gen. Virol.] and [54] be more appropriate than [25,26]?
Line 155-157:
Please recheck reference [21]. Not all 24 samples were successfully inoculated.
Line 157-159:
Please recheck reference [36]. HEV GT3 wbGER27 was inoculated but failed to replicate.
Line 167:
The previously mentioned study by Schemmerer et al. (2019, Viruses) showed that patient isolates would replicate to viral loads of ~109 HEV RNA copies/mL. Therefore I would restrain from stating that HEV from patient isolates generally replicate at low levels.
Line 209:
The reader might benefit if the recent reports were referenced here ([5,6]).
3.1 Hepatoma cell lines
In this section, the authors frequently mention the cell line A549 together with PLC/PRF/5 which is fine. However, for the reader not familiar with HEV cell culture systems, the authors may clarify at the first mention in this section that A549 is not a hepatoma cell line but a lung carcinoma cell line.
In addition, the authors could consider to mention that subclones of cell lines A549 and PLC/PRF/5 support HEV replication with different efficiencies (Shiota et al. 2015, Microbiol. Immunol. and Schemmerer et al. 2016, Viruses).
Line 295:
Although only few case reports are available (Anty & Ollier et al. 2012, J. Clin. Virol., Tabatabai & Wenzel et al., 2014 J. Clin. Virol. and Bouthry et al 2018, Emerg. Infect. Dis.), it seems that HEV GT3 is not dangerous for pregnant women as opposed to HEV GT1. This is also supported by the work of Gouilly et al. [69].
Line 267-268:
There are contradictory reports on whether HEV replicates better in A549 or PLC/PRF/5. It seems that A549 may generate adequate viral loads faster, but PLC/PRF/5 tend to generate higher viral loads at later stages. Moreover, Shukla & Nguyen et al. (2011, Proc. Natl. Acad. Sci.) found that Kernow-C1 replicated best in HepG2/C3A. Therefore, I would refrain from stating that HEV replicates more efficiently in A549.
Line 366-368:
The authors assume that HLCs, which can be cultured for up to one month, are suitable for studying chronic virus infection. However, several studies defined HEV infection as chronic when HEV RNA was detectable for >6 months in serum. Wouldn’t immortalised cell lines therefore be better suited to mimick chronic infections?
Line 444-446:
Would the authors consider it important that the HEV strain used in this study solely replicated in 3D-cultures but failed to replicate in monolayers?
Conclusion
In line 396-404, the authors nicely described the controversy which can be caused by cell culture-adapted HEV strains carrying insertions in the hypervariable region. This important aspect should be revisited in the conclusion.
Table captions are missing.
Spellchecking:
Line 101, 102 and 103:
PLC/PRF-5 should be spelled PLC/PRF/5.
Line 271:
PLCPRF5 should be spelled PLC/PRF/5.
Panels of Figure 2 should be labeled with A and B instead of 1) and 2).
Reviewer 3 Report
In the manuscript, the author summarizes the current knowledge regarding cell culture models of Hepatitis E Virus (HEV). The manuscript is well written and provides a nice overview about HEV and its in vitro systems. The tables and figures provide all information necessary. I don’t have any major critics. Reference number 7 about HEV antivirals could be updated with a recent review by Kinast et al. 2019 published in the HEV virus topic issue.
Round 2
Reviewer 1 Report
The authors changed the manuscript accordingly. However, in my previous comment, I think that the authors used too much space for the polarized hepatocytes cell culture models, and I suggest the authors to consolidate this section. The authors still did not consolidate this part and argue that they still want to keep it because of some points they proposed, some of which are even not published yet. Actually, I never ask them to delete this section, and I suggest that they should consolidate this section. The argument of the authors to my point is not convincing, and I cannot be convinced if the authors just present their points like "We do believed xxxx"(in the point-by-point letter).